# Efficacy of Kinematic Parameters for Assessment of Temporomandibular Joint Function and Disfunction: A Systematic Review and Meta-Analysis

**DOI:** 10.3390/bioengineering9070269

**Published:** 2022-06-22

**Authors:** Alessandra Scolaro, Shahnawaz Khijmatgar, Pooja Mali Rai, Francesca Falsarone, Francesca Alicchio, Arianna Mosca, Christian Greco, Massimo Del Fabbro, Gianluca Martino Tartaglia

**Affiliations:** 1Department of Biomedical, Surgical and Dental Sciences, University of Milan, 20122 Milan, Italy; scolaro.al@gmail.com (A.S.); khijmatgar@gmail.com (S.K.); poojarai93@gmail.com (P.M.R.); massimo.delfabbro@unimi.it (M.D.F.); 2Fondazione IRCCS Cà Granda, Ospedale Maggiore Policlinico, 20122 Milan, Italy; francescafalsarone7@gmail.com (F.F.); f.alicchio@gmail.com (F.A.); ari.mosca97@gmail.com (A.M.); 3Azienda Sanitaria dell’Alto Adige, Merano Hospital, 39100 Bolzano, Italy; christian.greco@sabes.it; 4IRCC Orthopaedic Institute Galeazzi, 20161 Milan, Italy

**Keywords:** temporomandibular joint, TMJ dysfunction, kinematics, motion analysis, temporomandibular joint disorders

## Abstract

The aim of this review was to answer the following PICO question: “Do TMJ kinematic parameters (intervention and comparison) show efficacy for assessment of mandibular function (Outcome) both in asymptomatic and TMD subjects? (Population)”. PubMed, Scopus, Web of Science, Embase, Central databases were searched. The inclusion criteria were (1) performed on human, (2) English only, (3) on healthy, symptomatic or surgically altered TMJ, (4) measured dynamic kinematics of mandible or TMJ (5) with six degrees of freedom. To assess the Risk of Bias, the Joanna Briggs Institute tool for non-randomised clinical studies was employed. A pairwise meta-analysis was carried out using STATA v.17.0 (Stata). The heterogeneity was estimated using the Q value and the inconsistency index. Ninety-two articles were included in qualitative synthesis, nine studies in quantitative synthesis. The condylar inclination was significantly increased in female (effect size 0.03°, 95% CI: −0.06, 0.12, *p* = 0.00). Maximum mouth opening (MMO) was increased significantly in female population in comparison with males (effect size 0.65 millimetres (0.36, 1.66). Incisor displacement at MMO showed higher values for control groups compared with TMD subjects (overall effect size 0.16 millimetres (−0.37, 0.69). Evidence is still needed, considering the great variety of devices and parameters used for arthrokinematics. The present study suggests standardising outcomes, design, and population of the future studies in order to obtain more reliable and repeatable values.

## 1. Introduction

The temporomandibular joint is a bilateral synovial joint between the temporal bone and the mandible, that function as one unit. Since the TMJ is connected to the mandible, the right and left joints must function together and therefore are not independent of each other. TMJ represents the only mobile joint in the skull. Mandibular stability and movements are essential to perform normal jaw function like biting, chewing, swallowing, and speech [1]. Loss of TMJ function, reduced coordination and pain are factors that determine social dysfunction and discomfort [2]. 

Temporomandibular disorders (TMD) are a group of heterogeneous clinical situations with multifactorial aetiology which can be related with musculoskeletal pain, joint noise and functional problems. TMD are a significant public health problem, and they represent one of the most common musculoskeletal conditions resulting in pain and disability [3] which mainly affects women. The prevalence of TMD is thought to be greater than 5–12% of the population. Regarding the symptomatology there is a peak occurrence between 20 and 40 years of age [4]. Temporomandibular joint disorders are classified in inflammatory and non-inflammatory pathologies which include disc displacement with reduction, disc displacement without reduction, structural incompatibility, adherence/adhesion, ankylosis, capsulitis, synovitis, retrodiscitis, dislocation, and osteoarthritis [3,5,6,7]. Chronic disc displacement and osteoarthritis are two most frequent TMJ disorders resulting in inflammatory and degenerative process of the joint structures [8]. TMD diagnosis is generally entrusted in physical and clinical examination findings and imaging, and more recently in kinematic analysis which is connected to function.

The treatment of muscle-related pain in TMD commonly consisted of occlusal splint devices, behavioural therapies and other conservative non-pharmacological approaches such as manual therapy, laser therapy, transcutaneous electrical nerve stimulation (TENS) and dry needling. Conservative and physical therapies are recommended for the initial treatment of TMD [9,10,11].

Mandibular kinematics measurements are used for clinical purpose such as study of dynamic occlusion in prosthodontic restorations, orthodontic and gnathological therapies, and for screening of temporomandibular dysfunction [12]. The correlation between TMD and mandibular kinematics has been described in literature: several studies evaluated TMJ function using various parameters such as condylar trajectories, incisal trajectories, mandibular rotation and translation, hinge axis, finite helical axis, intra-articular joint space and mastication cycle data in healthy and affected subjects [13]. However, there is still not enough scientific evidence to establish the physiological and pathological ranges of the kinematic parameters in order to obtain a strong efficacy in using this diagnostic tool.

Mandibular kinematics techniques can be classified in four categories: mechanical linkage systems, magnetic tracking systems, video motion analysis and radiographic tracking [14].

TMD are closely connected to mandibular movement and function, for that reason could be important to have an early and objective diagnosis. Therefore, the aim of this systematic review and meta-analysis is to demonstrate if and which kinematic parameters are useful to diagnose musculoskeletal jaw’s disorders, in order to assess clinical efficacy of this diagnostic tool for treatment planning. The outcome of this study may be useful also to lead future research studies on TMJ kinematic and its correlation to TMD.

## 2. Materials and Methods

This review followed the guidelines of the preferred reporting items for systematic reviews (PRISMA) [15]. The goal of this meta-analysis was to assess the efficacy of TMJ kinematics parameters for the early diagnosis of TMJ disorders, which is a controversial current clinical topic. The review was registered on the PROSPERO with a registration reference number 313476.

Participants/Population: Patients who are affected by temporomandibular disorders diagnosed according to diagnostic criteria for temporomandibular disorders (DC/TMD) for clinical and research applications, in particular myofascial pain, anterior disk displacement with or without joint noises, arthralgia, osteoarthritis and osteoarthrosis for at least three months. No particular limitations about patients age.

Intervention: TMD conservative treatment and non-surgical intervention. There were no restrictions of the duration of intervention.

Comparator (s): Asymptomatic subjects with different types of occlusion. No limitations about subjects age.

Outcome(s): Maximum mouth opening is distance between the incisal edge of the maxillary central incisors to the incisal edge of the mandibular central incisors at the midline when the mouth is open as wide as possible. Incisor displacement at MMO (IP-MMO) is the distance between the incisor point from the medial sagittal plane during maximum mouth opening. Condylar inclination represents the angle of the condyle translating down the condylar eminence as the mandible moves. Horizontal and sagittal angles are decomposition of the mandibular trajectory on the three planes of space.

Time: follow up at 6 mouths.

### 2.1. Eligibility Criteria

The eligibility criteria were planned considering the following population, intervention, comparison and outcome (PICO) question: “Do TMJ kinematic parameters (intervention and comparison) show efficacy for assessment of mandibular function (Outcome) both in asymptomatic and TMD subjects? (Population)”. We only included studies performed on humans and written in English. Moreover, opinion-based papers, perspectives, conference papers, book chapters, case reports and abstract were excluded. We also considered the cross-references of the included articles and reviews. The studies that met the following criteria: focused on healthy, symptomatic or surgically altered TMJ, measured dynamic kinematic of mandible or TMJ with six degrees of freedom were all included for qualitative and if found eligible quantitative analysis.

### 2.2. Search Strategy

A search for articles in the following electronic databases was carried out: PubMed, Scopus, Web of Science, Embase, Central. The databases were searched up until 2 January 2022. The key words used were: “temporomandibular joint” OR “TMJ” OR “mandible” OR “temporomandibular joint disorder” OR “TMD” AND “kinematics” OR “jaw motion” OR “motion analysis” OR “biomechanics”. A mix of medical subject headings and free key words were used. Only the studies published in English were included. The articles were manually checked for relevant articles to be included in the review. Once duplicates were removed, titles and abstracts were screened PRISMA flowchart was used to summarise the selection process (Figure 1).

### 2.3. Selection Process

Search strategy was implemented and tested by three researchers independently (GMT, AS and CG). All articles and abstract which met the selection criteria were included in the review. Three reviewers screened all the titles and abstract independently, those that seemed suitable were included in the full text review. When the information provided in the abstract and title were inadequate to determine eligibility, articles were included in full text review. Any disagreement among the reviewers was resolved by comparison discussion. At least two studies have to be considered to perform the meta-analysis, even if studies can be meaningfully pooled and provided their results are sufficiently ‘similar’ [13].

### 2.4. Data Collection Process

Data from the included articles were extracted by three independent reviewers and organised into a data sheet using Microsoft Excel 2013 (Microsoft Corp, Remond, WA, USA, https://www.microsoft.com/en-in/, accessed on 10 April 2022). A set of standard variables included was validated by discussing with all the reviewers. The validation was done based on the outcomes included in the previous literature used to assess the TMJ kinematics. The following items were extracted whenever possible: authors, year, country, journal, sponsor, study design, number, mean age and gender of the participants, number of test group, number of control group, TMJ condition, diagnostic tool, kinematic parameters analysed, mean values, number of samples in each group and standard deviation for groups. For each parameter, unit of measurement was specified. Data such as gender, study design, diagnostic tool, different TMD conditions for respective outcomes were used as sub-group analysis.

### 2.5. Data Items

The results of each study were recorded. Condylar trajectories, incisal trajectories, kinematic hinge axis, finite helical axis, intra-articular space joint, mastication cycle data and bennet angle were the outcomes of the analysed studies. All the outcomes were screened and recorded to assess if a comparison will be possible [16].

The main outcomes of the quantitative synthesis were maximum mouth opening (MMO) (degree and mm), Condylar inclination (degrees), horizontal (degrees) and sagittal angle (degrees) and linear displacement of the interincisal point at maximum mouth opening (IP)-MMO (mm) which are five parameters that represent an index of TMJ movement three-dimensionally. If different units of measure were identified, for the same parameter, we excluded them from quantitative synthesis and reported them in qualitative synthesis.

### 2.6. Quality and Risk of Bias Assessment

The risk of bias assessment of the included NRTCs was performed by three independent reviewers. The JBI—the Joanna Briggs Institute tool was used to assess the methodological quality of case control studies, cohort studies, cross sectional studies and case series. Each study was assessed according to predefined criteria. The JBI model of evidence-based healthcare conceptualises evidence-based practice as clinical decision-making that considers the best available evidence; the context in which care is delivered; client preference; and the professional judgement of the health professional [17]. The strength of this model is feasibility (setting realistic outcomes by knowing the right intervention and cost effectiveness), appropriateness (establish a best fit model for intervention), meaningfulness (establishing the experience) and effectiveness (scale at which intervention achieves the intended results and outcomes) [18]. The assessment tool was developed using a scored checklist of quality assessment questions. Each question was attributed a score of 1 or 0 based on whether the question in a given paper was clearly addressed or not addressed, respectively. The total scores were summed up. Studies with scores of 0–5, 5–6, and 7–11 points were considered low, moderate, and high-quality studies, respectively. Discussion among reviewers were used to solve any discrepancies in score.

### 2.7. Outcome Variables

Condylar inclination represents the angle of the condyle translating down the condylar eminence as the mandible moves. In this meta-analysis, the angle is evaluated during protrusive movement of the mandible and is measured in degrees.

The maximum mouth opening (MMO) has been defined as “the greatest distance between the incisal edge of the maxillary central incisors to the incisal edge of the mandibular central incisors at the midline when the mouth is open as wide as possible” [19,20]. MMO can be measured as a linear distance in millimetres or as an angle in degree. This meta-analysis compares MMO measured in millimetres.

Incisor displacement at MMO (IP-MMO) is the distance between the incisor point from the medial sagittal plane during maximum mouth opening, the unit of measure is millimetre.

Horizontal and sagittal angles are the indexes of mandibular movement which could be discomposed in three dimensions of space. These parameters are measures in degrees.

### 2.8. Measures of Treatment Effect

For continuous outcomes, we pooled data with the mean difference (MD), or standardised mean difference (SMD) if different measures were used to assess the same outcome. If the data are given for each patient, then we used the data to calculate the mean and sd using STATA software v17.0. The mean and sd were collected as mentioned in the published papers and there was no conversion of the values and the units of measure for pairwise comparison. If there were different levels of measure, then we would collect data in categorised form and use it in subgroup analysis.

MMO was analysed using as a reference point the value described by Travell [19] who measured 53 mm for women and 59 mm for men. Condylar inclination varies between 0° and 60°. The greatest frequency is around 40° to 50° [21]. To compare linear incisor displacement at MMO, values reported by Rieder [22], Ferrario [23] and Tsolka [24] were considered as the reference point.

### 2.9. Synthesis Methods

A meta-analysis was carried out only when there were at least two studies of similar comparisons reporting the same outcomes and unit of measure. A pairwise meta-analysis was carried out using STATA v.17.0 (Stata). The heterogeneity between each study was estimated using the Q value and the inconsistency index (I2 test). If the I^2^ is ≤50%, it suggests that there is negligible statistical heterogenicity and the fixed effects model will be employed. If the I^2^ is >50% we explored sources of heterogenicity by subgroup analysis and meta regression. If there was no clinical heterogenicity, the random effects model was used to perform the meta-analysis.

## 3. Results

### 3.1. Study Selection and Description

The initial research identified 28619 titles; 10017 remained after duplicates removal. Out of the remaining titles, 184 articles were considered potentially eligible and after a full text revision, verification of inclusion and exclusion criteria was carried out. Sixty-seven studies were excluded because the topic was not related to the review, two were studies on animals, seven on robotic actuators, one did not respect the six degree of freedom and four were systematic reviews. At the end of this stage, 92 were selected for qualitative synthesis. Eventually after outcomes assessment, eight comparative studies were selected for meta-analysis (Flowchart, Figure 1). The PICO assessment and the main findings are reported in Table 1.

### 3.2. Bias Risk Assessment

JBI Quality sores were collected and reported in an excel table. The JBI tool of the Case control studies ranging from 5 to 9. 7 were considered with moderate risk of bias, 10 studies showed high risk of bias. Cohort studies ranging from 4 to 9. 4 were considered moderate risk of bias, 10 showed high risk of bias. Cross sectional studies ranged from 5 to 7 with moderate quality. Case series studies raged from 1 to 8, all the studies showed high risk of bias.

#### 3.2.1. Qualitative Synthesis

The results of 92 articles were compared to evaluate the kinematic parameters useful to the assessment of TMJ function. Each study considered more than one parameter. A total of 58 studies measured condylar trajectories, 10 incisal trajectories, 8 mastication cycle data, 7 intra-articular joint space, 8 finite helical axis, 15 mandibular rotation and translation, 15 hinge axis and 1 muscular length.

All studies included were non-randomised clinical trials: 17 were case control studies, 14 were cohort studies, 59 were case series and 2 were cross sectional studies, involving a total of 3342 participants, of which 1127 were males and 2030 females, there were 2752 tests and 590 controls subjects. Among participants the following were reported: 813 with TMD, 201 with malocclusion, 204 with surgical intervention, 30 with condyle fracture, 8 with zygomatic fracture, 8 with cerebral palsy, 16 with Trigeminal neuropathic pain and 2062 asymptomatic subjects.

Among the included studies, 43 reported their funding source: 4 received funding from Research institutes, 9 from foundations, 14 from universities and 16 from Grants. The remaining studies did not declare any funding source as reported in the characteristics table (Appendix A).

#### 3.2.2. Quantitative Synthesis

A meta-analysis was performed considering the following outcomes: condylar inclination, maximum mouth opening (MMO), linear incisal displacement at MMO (IP-MMO), and horizontal and sagittal angle, which are mandibular motion indices at maximum opening. For each parameter the mean and standard deviation were recorded. The studies included for quantitative synthesis were Wieckiewicz et al. 2014; Baqaien et al. 2007; Reichender et al. 2013; Ferrario et al. 2005; Gallo et al. 1997; Mapelli 2016; Ugolini 2018; Mapelli 2016; Ugolini 2017.

### 3.3. Condylar Inclination

In three studies [25,26,27], condylar inclinations were compared between female and male with degrees as unit of measure. All subjects assessed were asymptomatic. Quantitative results show higher values for female with statistically significant difference (95% CI, Overall Effect size in degree 0.03 [−0.06, 0.12], Heterogenicity: T2 = 0.02, I2 = 57,76%, H2 = 2.37, Test of ϴi = ϴj = Q (19) = 45.24, *p* = 0.00) (Figure 2). Considering the diagnostic tool, axiography shows similar values in males and females, instead ultrasonic tracking technique favours female, however no statistically significant difference was found in term of diagnostic tool. (Figure 3). A subgroup analysis regarding condylar inclination was performed: age analysis shows a negative correlation in adult group, condylar inclination decreased when age increased. The mouth opening analysis demonstrates that there is no correlation between condylar inclination and level of opening (*p* = 0.65) (Figure 4).

### 3.4. Maximum Mouth Opening

Regarding MMO, three studies [27,28,29] were used to perform meta-analysis. The studies included MMO outcome, the unit of measure reported was in millimetre (mm). Male and female were compared: the overall effect size was 0.65 mm (−0.36, 1.66) with 95% CI, heterogeneity is I^2^ = 90.12% which indicates the studies were highly heterogeneous in characteristics. The pooled effect has passed the line of no effect, hence there exists statistically significant difference between the male and female groups in MMO outcome. The results were favouring females who showed higher values of MMO (Figure 5). Since the heterogeneity is very high, the results should be interpreted with caution and more studies are required to reach a meaningful interpretation.

The subgroup analysis shows that there is significant difference between the type of studies (case control and cohort studies) included and MMO outcome (Figure 6). Children and adult did not have significant values although the effect size varied. The Galbraith Plot for MMO shows the heterogeneity of the analysed studies: two studies were identified as outlier (Figure 7). There is a publication bias in MMO outcome as illustrated from funnel plot (Figure 8).

### 3.5. IP-MMO

The linear incisor displacement at MMO was analysed in two studies [30,31]. As for MMO, the unit of measure reported was millimetre. The comparison between TMD subjects and control group (asymptomatic) showed no significantly higher values for control asymptomatic subjects than TMD subjects (*p* = 0.05), the overall effect size being 0.16 mm (−0.37, 0.69) (95% CI) (Figure 9, Figure 10 and Figure 11).

### 3.6. Horizontal Angle

The horizontal angle was analysed in two studies [30,31] and reported in degrees. Treatment group was compared to control group: subjects who were surgically treated showed lower values than the controls. However, no statistically significant difference was found in terms of angle variation between the two groups (CI 95%, I^2^ = 0%, H2 = 1.00, *p* = 0.56). As for IP-MMO, the overall effect size was 0.16 (−0.37, 0.69) (Figure 12, Figure 13 and Figure 14).

### 3.7. Sagittal Angle

The sagittal angle was assessed in three studies [30,31,32] and reported in degrees. TMD subjects were compared to asymptomatic ones: there was no statistically significant difference between the two groups in sagittal angle outcome favouring TMD which showed lower values (CI 95%, I^2^ = 78.89%, H2 = 4.74, *p* = 0.85). The overall effect size was −0.09 degrees [−0.99, 0.82] (Figure 15, Figure 16 and Figure 17).

## 4. Discussion

Mandibular kinematics plays an important role in the evaluation of temporomandibular function. This aspect affects many oral health areas which are connected with the occlusal balance [33] such as prosthodontics, orthodontics and gnathology. The assessment of mandibular movement through condylar rotations and translation represents a cornerstone for TMD diagnosis, in terms of identification and classifying the degree of severity [34]. The quantification of mandibular movements allows both static and dynamic analysis of TMJ, that helps in making informed decisions in clinical setting. This review highlights the use of many arthrokinematic parameters but the absence of an effective gold standard of reference that can help in clinical practice.

Several studies aimed to measure kinematic parameters in order to identify specific values for asymptomatic, TMD [35,36] or surgically treated [37,38,39] subjects, using different diagnostic tools. Therefore, we aimed to determine the efficacy of kinematic in mandibular function assessment in different categories of subjects. A meta-analysis was performed in order to compare the outcome of several studies: only parameters reported with the same unit of measure was analysed. In our review, it was feasible to report few kinematics outcomes such as maximum mouth opening (MMO), horizontal angle, IP-MMO, condylar inclination, sagittal angle, and coronal angle.

Based on the qualitative analysis, we found that very few studies analyse the same parameters with the same units of measure. For example, Condylar inclination was assessed in several papers which considered the angle during opening, laterotrusion and protrusion but only four studies compared the angle during mandible protrusion [24,25,26,40] and among that, only three studies [24,25,26] evaluated condylar inclination during 5 mm of protrusion, involving 380 participants: 171 males and 209 females. Condylar inclination was higher in females than males and in younger than older subjects. Although is important to point out that the three studies included in the meta-analysis showed high risk of bias.

MMO was measured in eleven studies [26,27,39,41,42,43,44,45]; among these, eight studies reported MMO in millimetres and eventually only three [26,27,28] studies were included in the meta-analysis because of homogeneity of the participants (189 of which 97 were males and 92 females). Moreover, in this case the studies included showed high risk of bias, so the results must be evaluated with caution.

Linear incisor displacement at MMO (IP-MMO) was reported in four studies [29,30,36,38]. Eventually two [29,30] studies were included in the meta-analysis because of the comparability of the subjects (total participants are 71 of which 17 were male and 54 females) with high risk of bias. The results of the meta-analysis indicate that females display significantly higher values of condylar inclination and maximum mouth opening than males. This means that, the MMO and condylar inclination among females is more pronounced than males in TMD. Asymptomatic subjects present increased values regarding IP-MMO and horizontal angle than exposed (surgically treated subjects or with malocclusion) or suffering from TMD.

However, several limitations existed; therefore, the results should be interpreted with caution. First, the large panel of the existing arthrokinematics parameters and units of measure adopted by the different studies, which make difficult to compare clinical data. Second, modifications in head posture and reference point could affect the movement path of the mandible. In the absence of standardisation, this may give rise to a considerable variability. Third, heterogeneities among studies were often observed, which may be caused by differences in study populations and the high number of diagnostic tools that could be used for each study’ purpose. Fourth, the study design: most studies are case series with a higher risk of bias in data interpretation.

This meta-analysis shows that TMJ kinematic parameters are useful for mandibular function assessment in clinical practice, but studies included demonstrated that there is a need for data standardisation and establish a gold standard as a reference point. The current studies have heterogeneity in terms of outcomes and variables included. To improve the methods of strategies, the authors suggest carrying out future well-designed case-control studies to improve the level of evidence. Finally, it would be good if future studies compared the various diagnostic methods using the same parameters and units of measurement in order to be able to compare more data more reliably.

## 5. Conclusions

We concluded that, there is high heterogeneity among the majority of the outcomes except condylar inclination. Considering this factor, the interpretation should be done with caution and take a meaning full message that, the kinematic parameters are an important and useful tool for TMJ evaluation, both to evaluate function and dysfunction. There is a need of well-designed case-control or cohort studies that can add further evidence. The present review suggests standardising outcomes, study design, type of diagnostic devices used to assess the kinametics and population selection in the future studies. This would improve and add reliable and repeatable values.

## 6. Recommendations

A well-designed case-control studies would be beneficial in order to increase the level of evidence on the clinical significance of kinematics;The literature in the present review demonstrated varying diagnostic tools and measurements for different outcomes. It would be beneficial if the different diagnostic tools are used along with a standard of care;The present review highlighted different outcome measures and it would benefit if the case-control studies with MMO outcome as standard method would be measured along with the different outcomes which can have similar dimensions or unit of measurement;There are varying groups compared in the studies highlighted within this review. Comparing the groups that include, the TMD and the pre-surgical, post-surgical would bridge the gap in the evidence;Majority of studies included in this review lack sample size calculations. Therefore, it is highly recommended to use appropriate sample size calculations for respective studies.

## Figures and Tables

**Figure 1 bioengineering-09-00269-f001:**
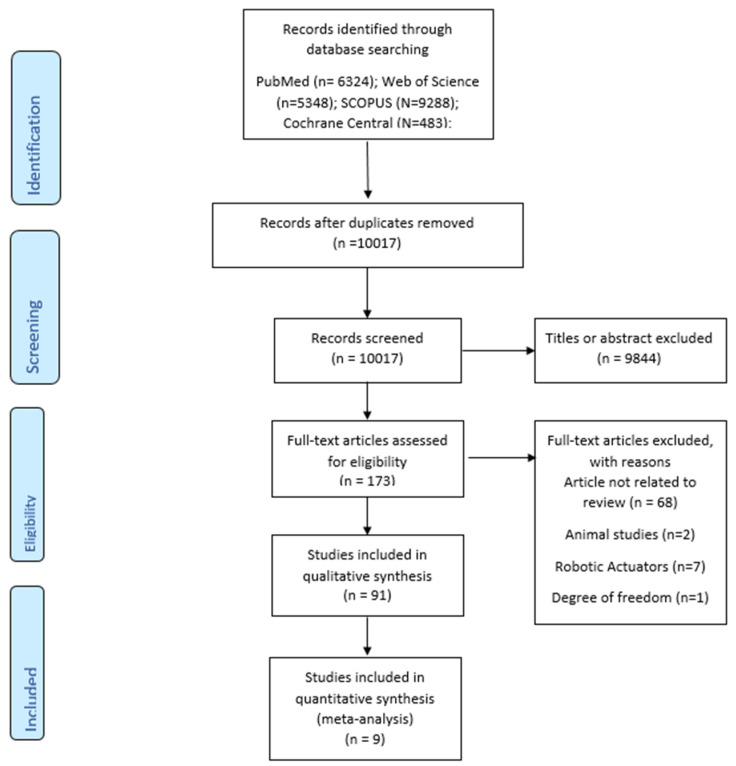
PRISMA flow chart.

**Figure 2 bioengineering-09-00269-f002:**
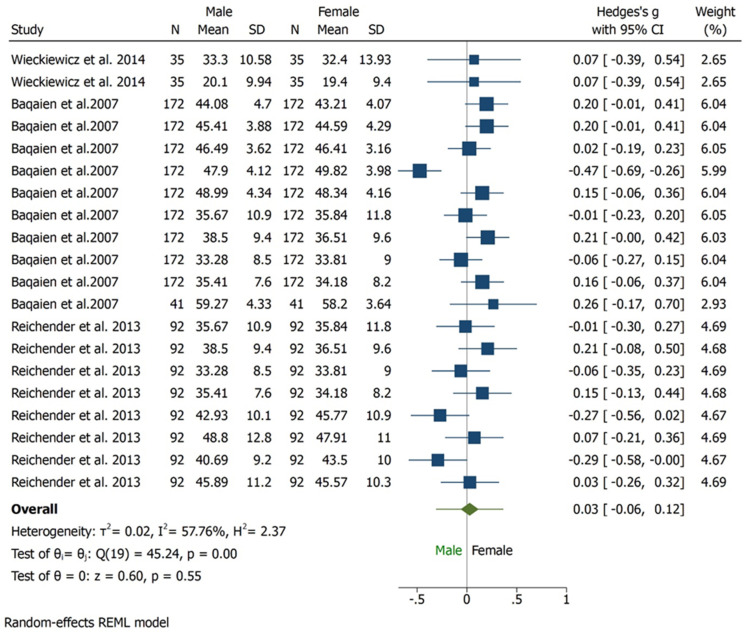
Forest plot for condylar inclination outcome.

**Figure 3 bioengineering-09-00269-f003:**
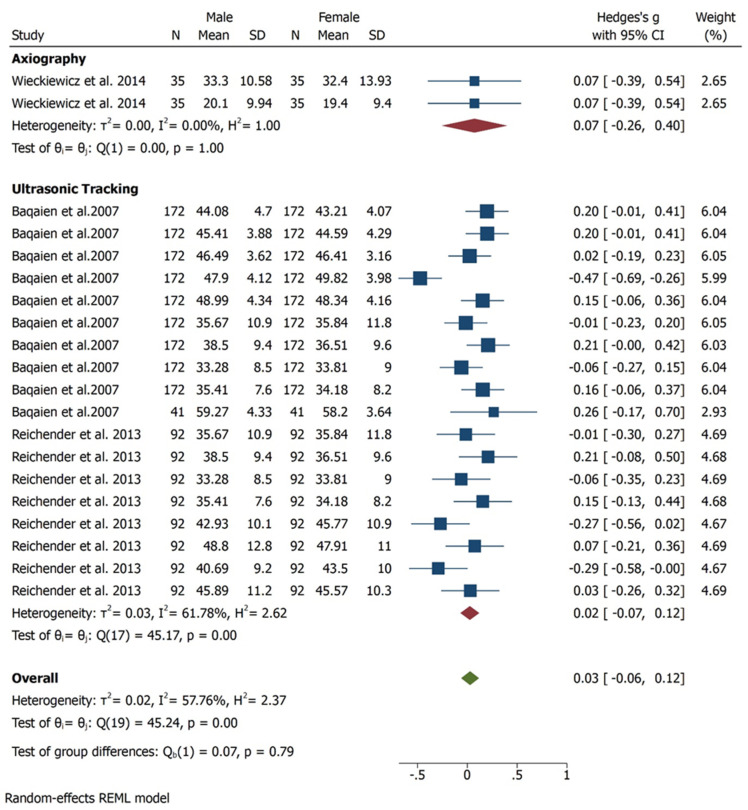
Forest plot for different method employed to estimate the condylar inclination.

**Figure 4 bioengineering-09-00269-f004:**
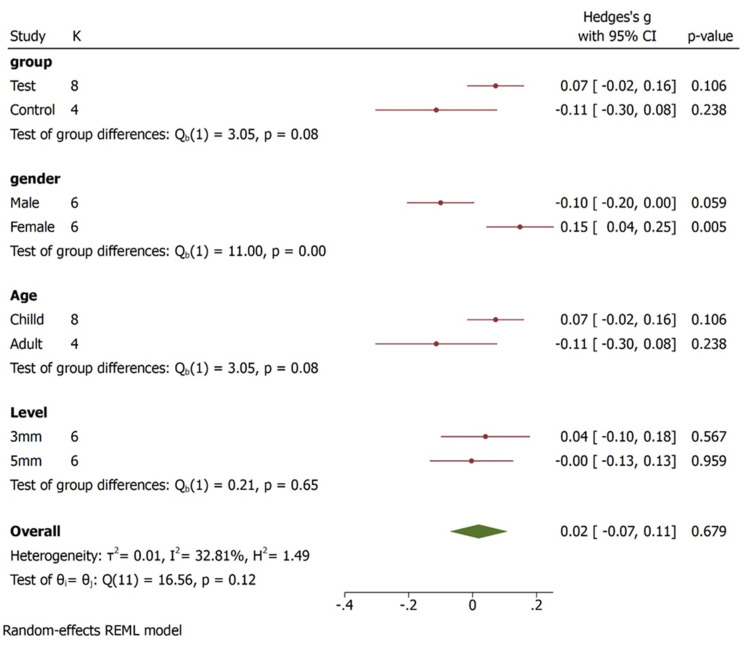
Condylar inclination subgroup analysis forest plot displays the study labels (id), the number of studies within each group (K), the plot of effect sizes and their CIs (plot), the values of effect sizes and their CIs (esci), and the p-values (*p*-value) of the corresponding significance tests. The between-group homogeneity test based on the Qb is reported for each subgroup analysis. For example, for subgroup analysis based on variable “Group”, there are two groups “test” and “control”. The test investigates whether the overall effect sizes corresponding to these two groups are the same. The results of this test are identical to those we would have obtained if we had specified subgroup (Group).

**Figure 5 bioengineering-09-00269-f005:**
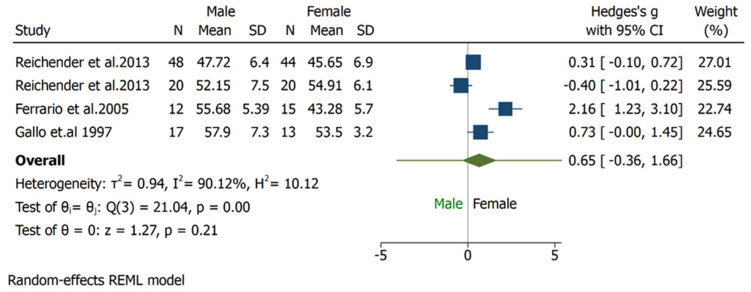
Forest plot for MMO outcome. The overall effect size was 0.65 (−0.36, 1.66). The highest weight was 27.01% with sample size N = 48 in males and N = 44 in females. The heterogeneity is I^2^ = 90.12% which indicates, the studies were found to be highly heterogeneous in characteristics. The pooled effect has passed the line of no effect, hence there exists statistically significant difference between the studies in MMO outcome favouring females.

**Figure 6 bioengineering-09-00269-f006:**
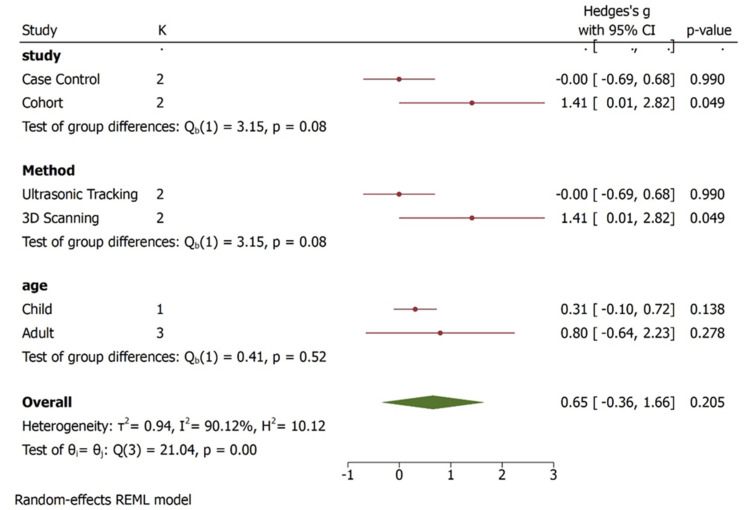
Subgroup analysis for MMO outcomes. The overall effect size was 0.65 (−0.36, 1.66). There is statistical significance between the type of studies included and its impact of results in MMO outcome. In addition, the method employed to record MMO outcome was statistically significant. Children and adult did not have significant values although the effect size varied i.e., child 0.31 (−1.10, 0.72) and adult 0.80 (−0.64, 2.23).

**Figure 7 bioengineering-09-00269-f007:**
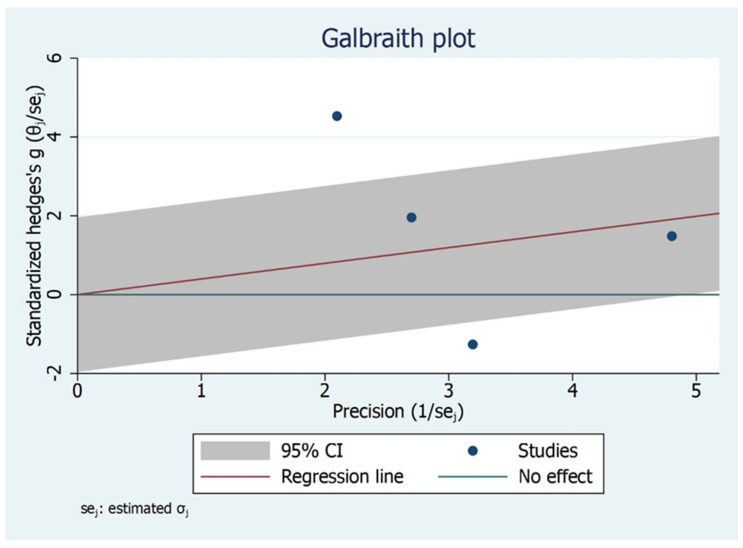
Galbraith Plot for MMO Outcome illustrating heterogeneity of the studies and detecting potential outliers. Two studies identified as an outlier.

**Figure 8 bioengineering-09-00269-f008:**
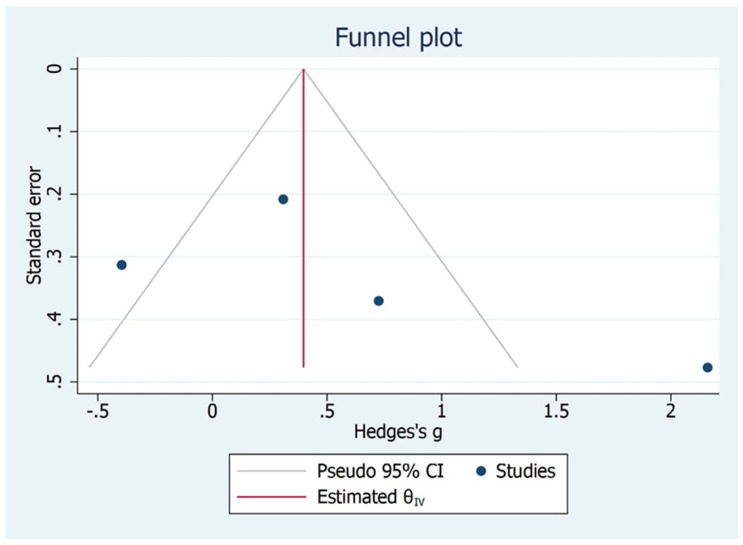
Publication bias for MMO outcome.

**Figure 9 bioengineering-09-00269-f009:**
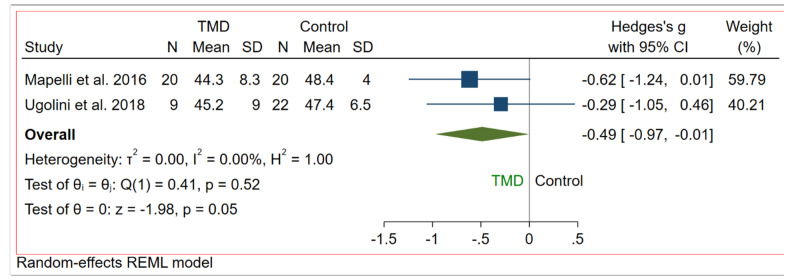
Forest plot for IP-MMO outcome.

**Figure 10 bioengineering-09-00269-f010:**
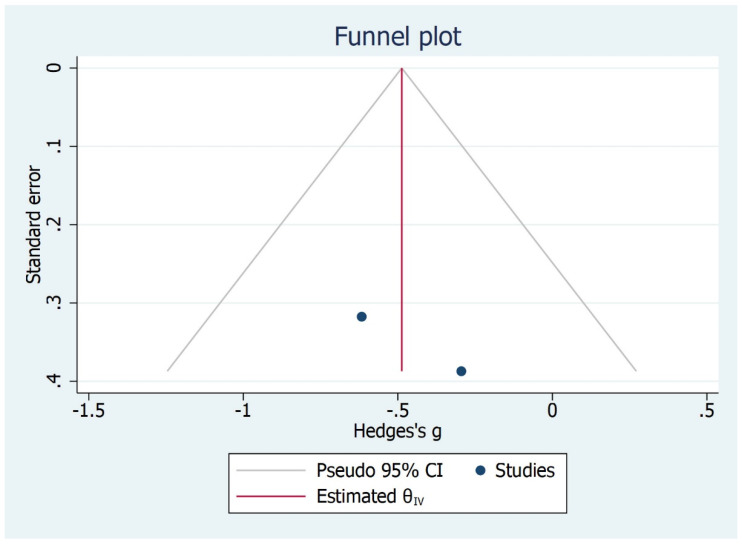
Publication Bias for IP-MMO outcome.

**Figure 11 bioengineering-09-00269-f011:**
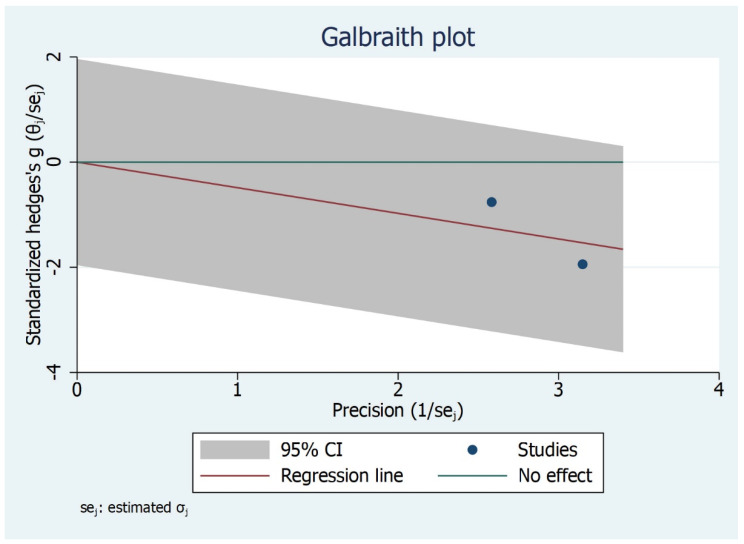
Galbraith plot for IP-MMO outcome.

**Figure 12 bioengineering-09-00269-f012:**
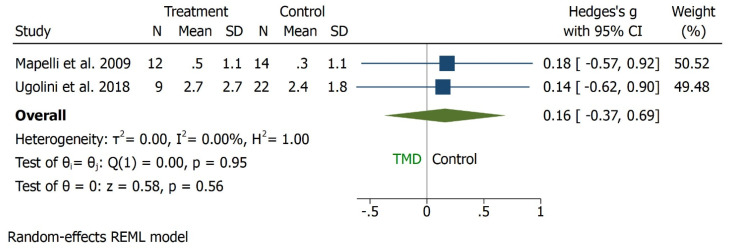
Forest plot for horizontal angle outcome.

**Figure 13 bioengineering-09-00269-f013:**
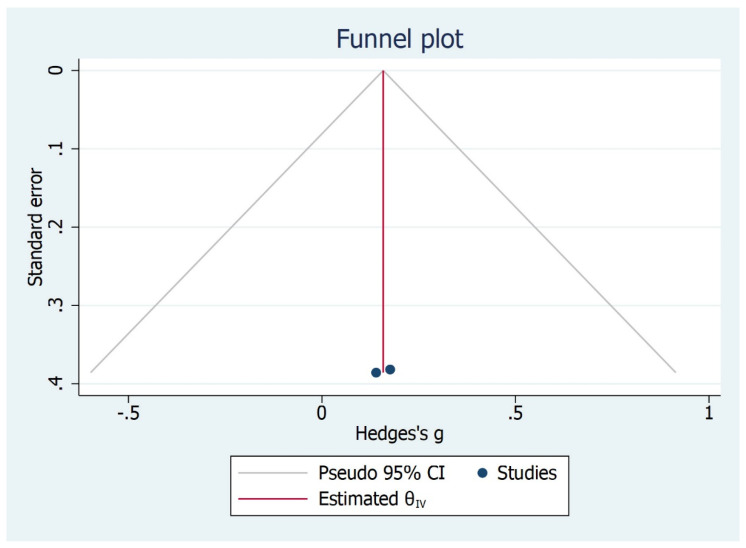
Publication bias for horizontal angle outcome.

**Figure 14 bioengineering-09-00269-f014:**
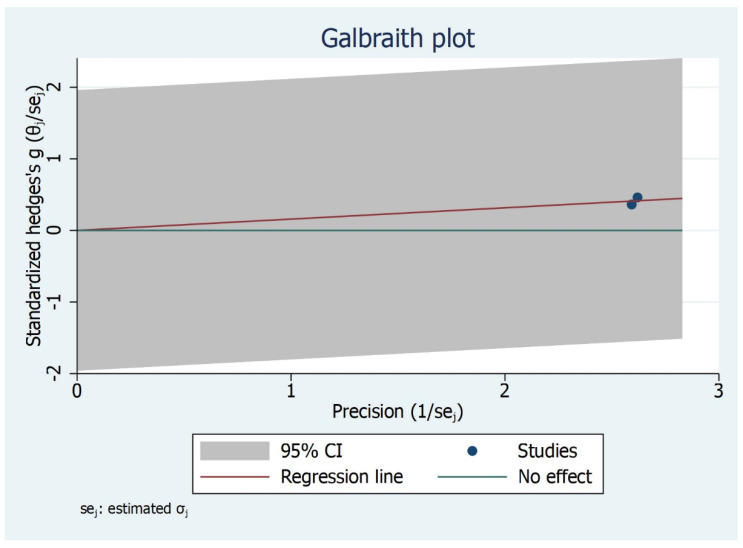
Galbraith plot for horizontal angle outcome.

**Figure 15 bioengineering-09-00269-f015:**
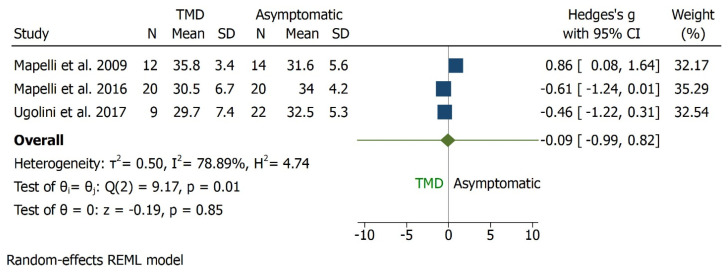
Forrest plot for sagittal angle outcome.

**Figure 16 bioengineering-09-00269-f016:**
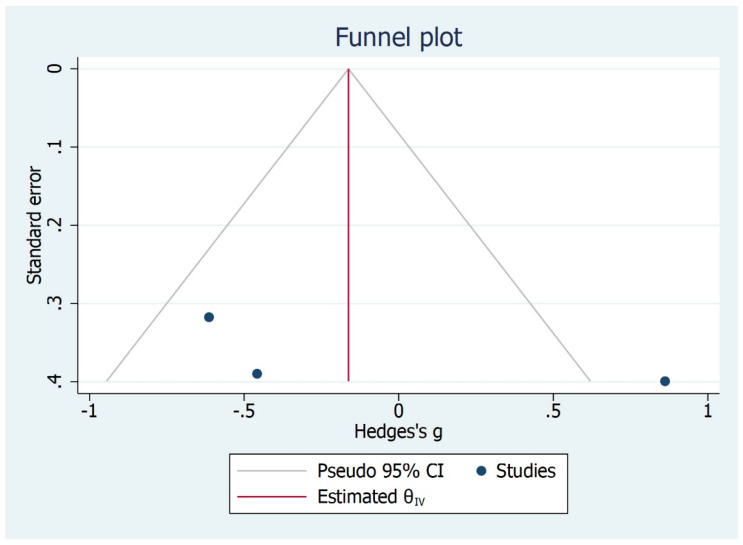
Funnel plot for sagittal angle outcome.

**Figure 17 bioengineering-09-00269-f017:**
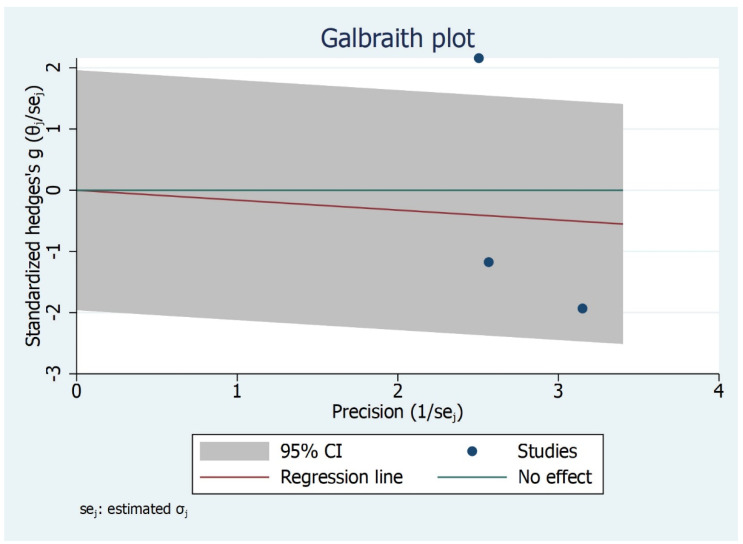
Galbraith plot for Sagittal angle outcome.

**Table 1 bioengineering-09-00269-t001:** PICO assessment and main findings.

Population/Participant	Diagnostic Criteria for Temporomandibular Disorders (DC/TMD) for Clinical and Research Applications, in Particular Myofascial Pain, Anterior Disk Displacement with or without Joint Noises, Arthralgia, Osteoarthritis, and Osteoarthrosis for at Least 3 Months.
Intervention	TMD conservative treatment and non-surgical intervention.
Comparison/Control	Asymptomatic subjects with different types of occlusion.
Outcome	Maximum mouth opening (MMO), Incisor displacement at MMO (IP-MMO), Condylar inclination, Horizontal and Sagittal angles.
Main finding	Heterogeneous variables and outcomes found to assess the TMJ kinematics. No gold standard outcome was found. No standardised control was found. Maximum mouth opening was commonly compared outcomes Different diagnostic methods were used for assessing TMD kinematics and among these methods could not be used as gold standard.

## Data Availability

All data is from previous published studies. Hence, not applicable.

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
