# Peer review of "Efficacy of Kinematic Parameters for Assessment of Temporomandibular Joint Function and Disfunction: A Systematic Review and Meta-Analysis"

_bioengineering, 2022, doi:10.3390/bioengineering9070269_

Round 1

Reviewer 1 Report

The paper entitled "Efficacy of Kinematic Parameters for Assessment of Temporomandibular Joint Function and Disfunction: A Systematic Review and Meta-Analysis" is an article of high clinical interest that analyzes the efficacy of different kinematic parameters assessing temporomandibular joint function, aiming to systematize and standardize the clinical diagnostic tool for treatment planning. This paper is undeniably an innovative research in relevant filed and make up the deficiency of previous analysis. A few minor revisions are given below.

INTRODUCTION

Introduction is well articulated and there are numerous references, most of which are relatively recent, with an acceptable level of scientific evidence and absolutely pertinent. The necessity and significance of this review is also clearly demonstrated.

I suggest to better explain the specific deficiency or limitation of present studies on TMJ kinematic parameters, highlighting the controversial point or vacancy in this field.

In addition, it is noted that you mentioned the mandibular kinematics techniques without logistical connection with the context or analyzing objects. Please complete this section to show its value or remove it when necessary.

Line 63/64 “the correlation between TMD and mandibular kinematics has been described in literature: several studies evaluated TMJ function using various parameters such as condylar trajectories”, However, there was no citation.

MATHERIALS AND METHODS 

Search and selection process are adequately detailed with an illustrative flow chart, especially emphasized the restriction on clinical investigation, which further adds to the data-oriented rigorousness as well as reliability. But there might be a bug in the number of studies included in meta-analysis, please check it again and modify it.

When deciding outcome data items, you took 5 parameters into consideration, please demonstrate the process or standard you select them and how do they contribute to the appraisal of TMJ kinematic function and dysfunction.

In terms of quality and risk of bias assessment you employed the JBI model to assess the methodological quality of 4 kinds of studies, I suggest a complementary explanation of the applicability of this model.

Meanwhile, it is noted that you employed MD and SMD to assess the same item of continuous outcomes, making the treatment effect more comparable. It would be much better if you add the methods how you convert and unify differently measured outcomes.

RESULTS

Results are highly conformed with above methods with abundant valid supporting data. The included studies are clearly classified, the gender and TMJ health situation of included samples are carefully counted, and the sample sizes are also considerable, contributing to a higher level of authority compared to previous similar analysis.

DISCUSSION AND CONCLUSION AND RECOMMENDATIONS

Discussion objectively summarized above results data, admitting high risk of bias of some included studies. However, you didn’t directly answer the question whether you find out efficient kinematic parameter in mandibular function assessment and if not, what’s the reason. Besides, it recommended to add some simple comments of how you regard the conclusion, whether you are satisfied with it, and if not, how would you improve the methods or strategies.

REFERENCES

It is noted that there is a reference article dating back to 1961, please check whether it’s applicable or outdated to the review.

Author Response

Dear Reviewer,

Please find the attached document. 

Thank you

Regards

Shahnawaz K

Reviewer 2 Report

Alessandra Scolaro, et al. reviewed the efficacy of kinematic parameters for the assessment of temporomandibular joint function and dysfunction.

This is a good review supported by enough references surveys and analysis. I just recommend the authors emphasize the discussion or introduction in terms of the research background and its significance.

Author Response

Dear Reviewer,

Please find that attached document for your review.

Thank you very much

Regards

Shahnawaz K

Reviewer 3 Report

Thank you for giving me this opportunity to review this paper. 

My comments are 

1. Authors stated the inclusion criteria were based on PICOS criteria.  However,  they did not mention details about each one. Authors should stated type of patients or Problem such arthrogenous TMDs or  myogenous TMDs , anterior disc displacements with reduction , arthragia, osteoarthritis etc should mentioned details about patient's age , type of diagnosis 

Also, type of interventions, it's mandatory to clarify what we're the interventions such as conservative treatment,  or surgical treatment such arthroplasty, arthrocentesis, arthroscopic etc .Additionally,  details about the signs and symptoms,  duration of TMD 

A similarly to outcomes , study design and time ( follow up). It's mandatory to clarify the kinematic parameters in details that analyzed and details for each parameters? (definition) 

2. Flow chart, is incomplete . Author included different studies in quantitative and qualitative assessment.  This is incorrect,  authors should included only studies that included in meta-analysis based on PICOS criteria. 

3. Authors included all type of studies even the case series studies? This is a problem in risk of bias assessment?

4. Authors compared between male and female in respect of condylar inclination,  only 3 studies included , authors used multiple reading for the same studies, this is incorrect analysis,  authors should used the same and specific parameters for all studies. Otherwise , the results will be inaccurate abs misleading. 

4. Most of analyses were done on few studies with inaccurate data. In this case , meta analysis is not appropriate and shouldn't be preformed and the yielded results will be incorrect and shouldn't be used as basis for decision making or drawing any strong conclusion. 

From methodological point of view,  we can do meta-analysis in this study and meta-analysis will be invalid. 

included studies had an extremely heterogeneity and are not appropriate to pooled it in one meta analysis.  

Authors preformed comprasions in respect of female to male , TMDs and asymptomatic and control vs treatment while they did not clarify any details in methodology. 

Summary : methodology is not clearly defined,  inclusion criteria is not completely defined,  meta-analysis is not valid, and there was  severely heterogeneity among studies that shouldn't be included in one studies. Thus there is a major fatal mistakes. That is why I recommend against acceptance 

Author Response

Dear Reviewers,

Please find the attachment of the document as requested with responses.

Thank you for reviewing the manuscript.

Regards

Shahnawaz K

Reviewer 4 Report

Dear Authors,

In my opinion, 28 references for a systematic review and meta-analysis are too few. The manuscript did not respect the author’s guidelines. The topic is very interesting, however, the text should be greatly improved and the manuscript should be copyedited by a native English speaker or copyediting service.

I suggest some revisions to improve this work.

Major revisions

Abstract Section

The abstract should be a total of about 200 words maximum. The abstract should be a single paragraph and should follow the style of structured abstracts, but without headings. Follow the Instructions for Authors, please

 (https://www.mdpi.com/journal/bioengineering/instructions#preparation)

Keywords

Please add “temporomandibular joint disorders”.

INTRODUCTION

  • “Temporomandibular joint is a bilateral synovial joint between the temporal bone and the mandible, that function as one unit. Since the TMJ is connected to the mandible, the right and left joints must function together and therefore are not independent of each other. TMJ represent the only mobile joint in the skull. Mandibular stability and movements are essential to perform normal jaw function like biting, chewing, swallowing and speech” Please add references (Okeson JP. Management of Temporomandibular Disorders and Occlusion, Missouri Elsevier Mosby, St. Louis, MI, USA, 7th edition, 2012).
  • “Temporomandibular joint disorders are classified in inflammatory and non-inflammatory pathologies which include disc displacement with reduction, disc displacement without reduction, structural incompatibility, adherence/adhesion, ankylosis, capsulitis, synovitis, retrodiscitis, dislocation and osteoarthritis”. Please refer to the TMD classification according to the Diagnostic Criteria for Temporomandibular disorders (DC/TMD) (Schiffman et al. Diagnostic Criteria for Temporomandibular Disorders (DC/TMD) for Clinical and Research Applications: recommendations of the International RDC/TMD Consortium Network* and Orofacial Pain Special Interest Group†. J Oral Facial Pain Headache. 2014;28(1):6-27. doi:10.11607/jop.1151).
  • Line 55: “structural incompatibility”. What do you mean?
  • In the Introduction Section, please refer also to the common treatments (You should cite: “Efficacy of rehabilitation on reducing pain in muscle-related temporomandibular disorders: A systematic review and meta-analysis of randomized controlled trials. doi:10.3233/BMR-210236; “Are occlusal splints effective in reducing myofascial pain in patients with muscle-related temporomandibular disorders? A randomized-controlled trial. doi:10.5606/tftrd.2021.6615; Oxygen-Ozone Therapy for Reducing Pro-Inflammatory Cytokines Serum Levels in Musculoskeletal and Temporomandibular Disorders: A Comprehensive Review. doi:10.3390/ijms23052528).
  • “The correlation between TMD and mandibular kinematics has been described in literature: several studies evaluated TMJ function using various parameters such as condylar trajectories, incisal trajectories, mandibular rotation and translation, hinge axis, finite helical axis, intra-articular joint space and mastication cycle data in healthy and affected subjects.” Please add references.

MATERIAL AND METHODS

- The Eligibility Criteria are not clearly reported. In particular, the Outcome is not well described.

- “Any clinical investigation evaluating TMJ kinematics concerning TMJ disorders was included”. Please correct the English grammar.

- The PRISMA Flow Chart is not correct (173-(68+2+7+1) ≠ 91). Moreover, the final number of included articles in missed. Please refer to the PRISMA 2020 statement (Page et al. The PRISMA 2020 statement: an updated guideline for reporting systematic reviews. BMJ 2021;372:n71. doi: 10.1136/bmj.n71).

- “Search strategy was implemented and tested by three researchers independently (GMT, AS and LX)”. Who is LX? Among the authors, nobody is called LX.

- “Any disagreement between the reviewers was resolved by comparison discussion”. The reviewers were more than two, so use “among” not “between”.

- “Quality and Risk of Bias Assessment” please add reference.

RESULTS

  • It is NOT clear how many articles were included in the review. At the beginning of the Results Section, please add the references of the articles included in the study. 92? 8? It seems that 92 articles were included, whereas 8 were included in the meta-analysis. However, results about the 92 papers were not reported.
  • Please add a Table reporting, Intervention, Comparison, Outcome measure, and Main findings.
  • - Line 256 and 286: “millimeter”
  • “Maximum Mouth Opening”. The section is not clear. Please, rewrite it.ù

DISCUSSION

- Lines 331-332-333-342-343-348-349-350-355-356. All the references must be reported in square brackets.

- “For example, Condylar inclination was assessed in several papers which considered the angle during opening, laterotrusion and protrusion but only four studies compared the angle during mandible protrusion (Hue et al., Reichender et al., Wieckiewicz et al.Baqaien et al.) and among that, only three studies [17–19] evaluated condylar inclination during 5mm of protrusion, involving 380 partici pants: 171 males and 209 females.”

- Authors should discuss the results and how they can be interpreted in perspective of previous studies and of the working hypotheses. The findings and their implications should be discussed in the broadest context possible and limitations of the work highlighted. This section should be improved, in order to compare the Results to the Scientific Literature, reporting the main finding of the work. Despite the great work done by the authors in reviewing the scientific literature, the clinical findings are not well described.

CONCLUSION

Conclusion Section is not clear. Grammar is not correct.

Author Response

Dear Reviewer,

Please find the attachment of the document as requested with responses.

Thank you

Regards

Shahnawaz

Round 2

Reviewer 3 Report

Authors addressed my comments appropriately